# A Comprehensive Survey on AI-based Methods for Patents

## Abstract

Recent advancements in Artificial Intelligence (AI) and machine learning have demonstrated transformative capabilities across diverse domains. This progress extends to the field of patent analysis and innovation, where AI-based tools present opportunities to streamline and enhance important tasks in the patent cycle such as classification, retrieval, and valuation prediction. This not only accelerates the efficiency of patent researchers and applicants but also opens new avenues for technological innovation and discovery. Our survey provides a comprehensive summary of recent AI tools in patent analysis from more than 41 papers from 26 venues between 2017 and 2024. Unlike existing surveys, we include methods that work for patent image and text data. Furthermore, we introduce a novel taxonomy for the categorization based on the tasks in the patent life cycle as well as the specifics of the AI methods. This interdisciplinary survey aims to serve as a resource for researchers and practitioners who are working at the intersection of AI and patent analysis as well as the patent offices that are aiming to build efficient patent systems.

## 1 Introduction

Recent progress in AI and machine learning has shown transformative capabilities across various domains including NLP, computer vision, and healthcare Devlin et al. (2019); Radford et al. (2019). The field of patents and technological innovation is not an exception. AI-based tools can streamline the complex patent related tasks such as classification, retrieval, and valuation prediction. For instance, for patent examination, patent offices often rely only on the examiner to judge whether a technology is innovative enough and thus, patentable. However, it is challenging for the human examiner to stay updated on various domains due to the exponential growth in technology and apply the knowledge during evaluation. This intersection of AI and patent processes can accelerate the efficiency of the patent systems—patent reviewers as well as applicants—and help in a faster technological innovation to benefit our society.

The patent application and granting process in the patent life cycle involves complex tasks that require significant human effort for both applicants and reviewers. To streamline these complex processes, AI can be helpful, particularly in patent classification, retrieval, and quality analysis Krestel et al. (2021). Patent classification can benefit from the AI-based multi-label classification tools for the hierarchical schemes: International Patent Classification (IPC) and the Cooperative Patent Classification Roudsari et al. (2022); Althammer et al. (2021). To evaluate novelty and avoid infringement, the patent retrieval task becomes important while filing or reviewing a new patent application. On the other hand, quality analysis also requires a substantial amount of effort. AI-based representation learning methods can be useful in both tasks Chung & Sohn (2020); Lin et al. (2018). Lastly, recent generative AI tools can generate accurate and technical language descriptions for patents and thus, are useful to optimize human resources and precision in patent writing Lee & Hsiang (2020b).

In the literature, there is a lack of recent surveys on AI tools for patent analysis. The most recent survey Krestel et al. (2021) does not cover the recent studies in this area. Moreover, it focuses heavily on text-based approaches. Our survey aims to bridge this gap by providing a comprehensive and detailed summary of existing AI methods in more than 40 papers that appeared in 26 different venues from 2017 to 2024 for patent analysis. We include recent AI tools both for images and text data in patent analysis. Moreover, we introduce a novel taxonomy to categorize these methods based on the relevant tasks and the nature of the methods.

Our taxonomy is unique in a few ways. First, our taxonomy is based on the AI tools and patent-related tasks. This provides an in-depth view of the methods being used in specific tasks. Additionally, it organizes the previous works based on the type of the methods (e.g., Neural networks, Pre-trained language models, etc.) and provides granularity. This will be also beneficial for future researchers who will aim to focus on a task-specific method. Also note that, earlier studies do not offer this type of taxonomy that directly narrows down to specific tasks and methods. Second, another uniqueness about this organization is that it captures the recent trends of using advanced methods (e.g., LLMs) which is missing in the existing survey.

## 1.1 Overview

**Search and inclusion criteria.** We have conducted our literature search using Google Scholar and Semantic Scholar, focusing on various categories of patent-related tasks and the application of AI methods. To align with the recent trends, we have limited our search to publications from 2017 to 2024. Our search criteria included various keywords such as 'patent', 'AI in patent', 'patent classification', 'patent tasks', 'patent retrieval', 'survey in patent', 'patent generation', 'patent quality analysis', and 'patent dataset'. This combination of search terms has yielded hundreds of patent-related research papers. We have excluded more than half of these papers after reviewing their titles and abstracts, as they have not met our criteria (e.g., they did not fall under any of the relevant categories). For instance, one retrieved paper titled 'Generating patent development maps for technology monitoring using semantic patent-topic analysis' is not relevant to this survey. After thorough scrutiny and reorganization, we have included more than 40 papers for the survey.

Figure 1 provides the hierarchical organization where the most important tasks and their related methods are presented. We organize the survey into the following sections: Section 2 provides background on both the relevant tasks in the patent life cycle as well as the patent datasets. Section 3 summarizes the methods for four individual tasks: patent classification, retrieval, quality analysis, and generation. These methods are further grouped based on their commonality. Finally, Section 4 provides a few important research directions including the use of generative AI and multimodal learning. Note, the frequently used AI methods in the papers covered by this survey are referred in Table 2.

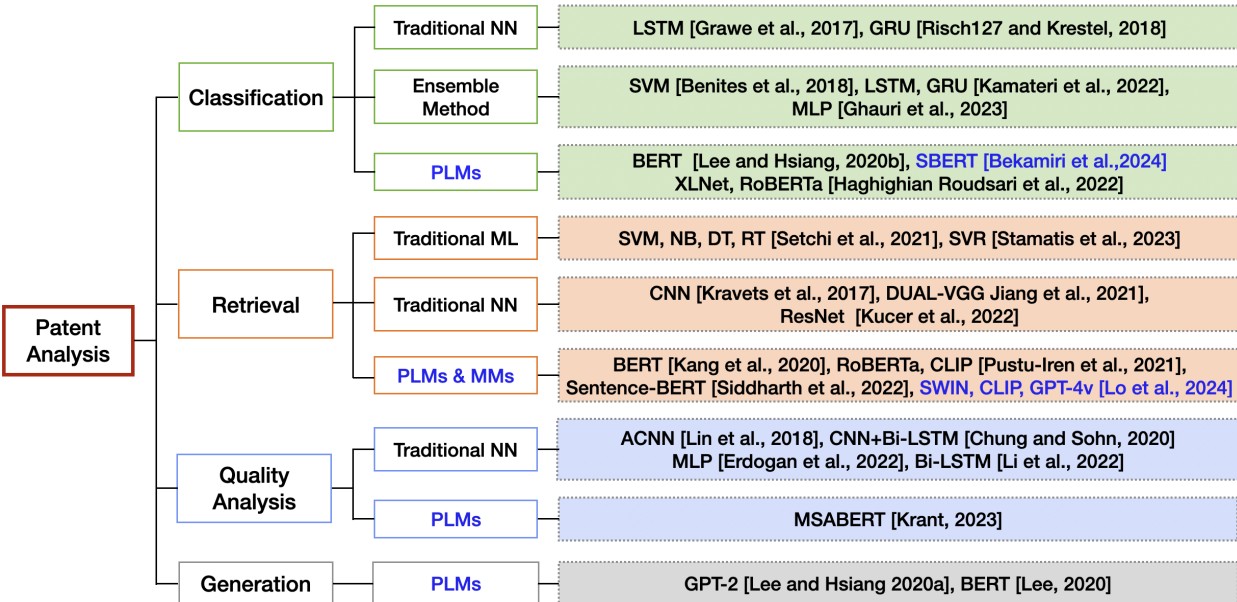

Figure 1: The schema of the main organization with a few examples of the methods in each patent related tasks. We summarize the methods for four individual tasks: patent classification, retrieval, quality analysis, and generation. "NN", "MMs", and "PLMs" denote neural network, multimodal models, and pre-trained language model respectively.

## 2 Background

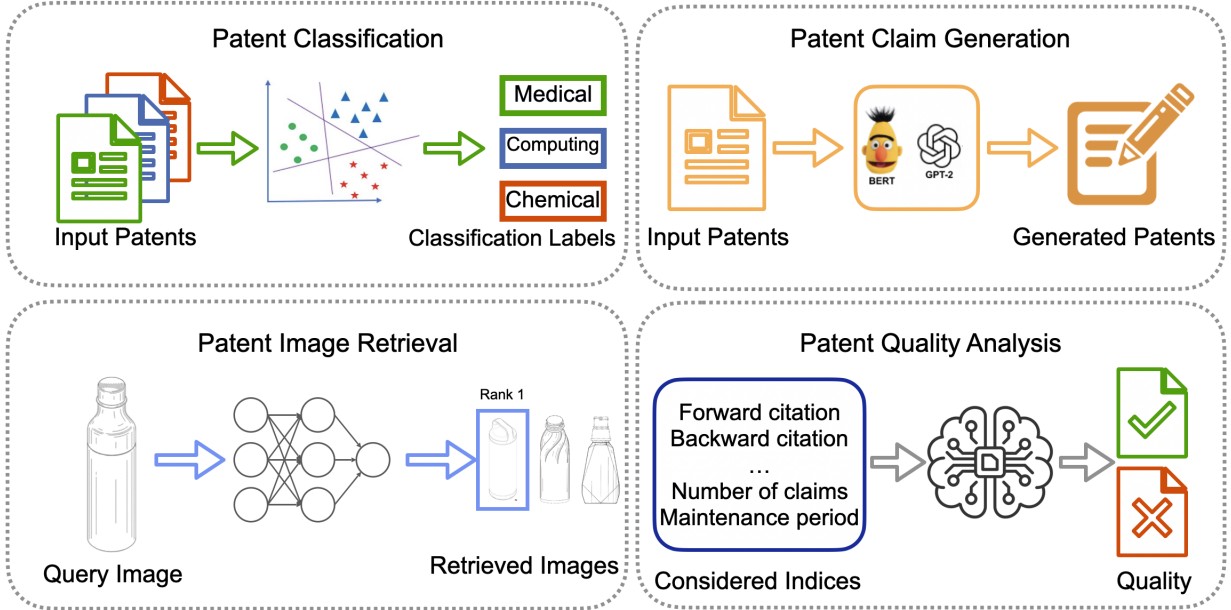

Figure 2: The overview of four major tasks of patent analysis. The patent retrieval task includes obtaining relevant patents (text and images). Please refer to the detailed descriptions of these tasks in Section 2.1.

A patent grants the owner or holder exclusive rights to an invention, and can be a novel product or a process that usually offers a unique method or technical solution. In exchange for this right, inventors must publicly disclose detailed information about their invention in a patent application[1]. The United States Patent and Trademark Office (USPTO[2]) issues three types of patents: utility, design, and plant. Utility patents protect the rights related to how the invention works or is used. It provides the entitlement to the functionality of a product. On the other hand, design patents protect the right of the look of an invention and are intended to safeguard the form of a product. A plant patent is issued to an inventor who has discovered and invented a unique variety of a plant and asexually reproduced it. In this work, we focus on utility and design patents considering the relevance of the AI tools on the tasks related to these categories. Here, we describe the relevant tasks, the datasets, and their sources.

### 2.1 Tasks

Patent application and granting processes are complex and involve many complex tasks that include both applicants and patent reviewers. The AI tools can help in simplifying these tasks. Here, we outline the most relevant tasks that can exploit AI-based methods.

#### 2.1.1 Patent Classification

Patent classification is an important and time-consuming task in the patent life cycle Grawe et al. (2017); Shalaby et al. (2018); Risch & Krestel (2018). This task involves a multi-label classification for patents where the classification scheme is hierarchical and a patent can get multiple labels. There are two widely used patent classification systems: the International Patent Classification (IPC) and the Cooperative Patent Classification (CPC). The IPC comprises 8 sections, 132 classes, 651 subclasses, 7590 groups, and 70788 subgroups in a hierarchical order (i.e., sections have classes and classes have subclasses, and so on). CPC is an expanded version of the IPC and is collaboratively administered by the European Patent Office (EPO)

---

[1]https://www.wipo.int/classifications/ipc/
[2]https://www.uspto.gov/

and the USPTO. It consists of around 250,000 classification entries and is divided into nine sections (A-H and Y), which are further broken down into classes, subclasses, groups, and subgroups[3]. Table 1 shows an example of CPC classification.

**Formulation.** Given patents as $(x_i, y_{ik}), i = 1, \ldots, N$, where $x_i$ are the features of the example patent $i$, $k \in C$, $C$ denotes the set of classes, $y_{ik} \in \{0, 1\}$ is a binary indicator of whether class $k$ is the correct classification for the example patent $i$, and $i$ can belong to more than one class in $C$. The goal is to predict $y_{ik}$.

**Challenges.** Patent classification is challenging due to its multi-class and multi-level nature. A single patent can be assigned multiple CPC/IPC codes, which makes the classification process complex. Additionally, patent documents have various sections such as titles, abstracts, and claims—each contains different information. Given the extensive length of these full-text patent documents, identifying the most relevant sections for classification also poses a significant challenge.

Table 1: An example of Cooperative Patent Classification (CPC) Scheme for the section A and its hierarchical categorization.

| Level | Code | Category |
|-------|------|----------|
| Section | A | Human Necessities |
| Class | A61 | Medical or Veterinary Science: Hygiene |
| Sub-class | A61B | Diagnosis: Surgery: Identification |
| Group | A61B5 | Measuring for diagnostic purposes; Identification of persons |
| Sub-group | A61B5/0006 | ECG or EEG signals |

### 2.1.2 Patent Retrieval

Patent Retrieval (PR) Kravets et al. (2017); Kang et al. (2020); Chen et al. (2020); Setchi et al. (2021) focuses on developing methods to effectively and efficiently retrieve relevant patent documents and images based on specific search queries. PR plays a crucial role in identifying new patents related to new inventions. It is essential for evaluating novelty of a patent as well as ensuring that it does not infringe on existing patents. Moreover, patent image retrieval can serve as a source of inspiration for design.

**Formulation.** Given a query patent as $q$ and a set of patents $X = \{x_i, \ldots, x_n\}$, where $x_q$ and $x_i$ are the features of the query and the patent $i$ in the set $X$. The goal is to compute a similarity score (e.g. cosine) $s(x_q, x_i)$ and return a set of patents $R(q) = \{x_j, \ldots, x_k\}$ based on top-k high similarities.

**Challenges.** Patent retrieval tasks involve both text and image retrieval where each have unique challenges. Text retrieval is complex due to the use of similar words to describe new inventions; an invention can be described using various synonyms and phrasings which make it difficult to retrieve crucial information for patent infringement analysis. On the other hand, image retrieval is particularly challenging due to the nature of the images involved, which are typically black and white sketches including numbers to describe the part of the inventions.

### 2.1.3 Quality analysis

Businesses have shown great interest in evaluating patent value due to its significant impact in generating revenue and investment Aristodemou (2021). Investors usually aim to predict the future value of a technological innovation from the target firm while making investment decisions. As a result, many companies hire professional patent analysts for quality analysis. This complex task demands substantial human effort as well as expertise in various domains Lin et al. (2018). The quality of a patent can be assessed using various measures, including the number of forward or backward citations, the number of claims, the grant lag, patent family size, the remaining lifetime of the patent Aristodemou (2021); Erdogan et al. (2022).

**Challenges.** The challenge in analyzing patent quality is the ambiguity of the metrics to quantify the quality of a patent. Commonly used measures for the quality analysis are the number of citations (both forward and

---

[3]https://www.cooperativepatentclassification.org/

backward), the number of claims, and the grant lag. However, the weight of each of these measures remain unclear. Moreover, analyzing these information to perform a comprehensive study is non-trivial.

### 2.1.4 Patent Generation

Patents usually have a considerable amount of written text which requires significant human resources. The patent generation task involves generating specific sections of a patent, such as abstract, independent claims, and dependent claims, based on instructions for an AI tool. Patent documents require precise and technical language to accurately describe the invention and its claims Risch et al. (2021). AI-assisted patent generation will help to automate the drafting process which involves time, effort, and legal requirements. This will also reduce the amount of patent attorney time which will be a substantial cost saver.

**Formulation.** Given the patent $x_i$, where $x_i$ are the features of the example patent $i$. $x_i$ can be features constructed from the title, abstract, or any other part of the patent, the output $y_i$ can be another part of the patent (e.g., abstract, the first claim). The generation function $G$ can be denoted as $y_i = G(x_i; \theta)$, where $\theta$ is the parameter of the generation model $G$. The goal is to generate $y_i$ by learning $\theta$, or inferring from a pre-trained model with learned $\theta$.

**Challenges.** Though the patent document has certain structure, one major challenge is to evaluate the dependency between the specific parts of the patent. This dependency could be exploited in the generation process. For instance, one part (e.g., abstract, claims) can be used as an input and a context in a generative model (e.g., a PLM) to generate a different part of the patent part. Additionally, it becomes non-trivial to construct effective instructions or prompts that guide the generation process. The generation also brings the question of evaluation of the generated content or text, i.e., how to appropriately judge whether the generated patent content is desired or not.

Table 2: Popular AI methods in the literature. We use the acronyms frequently in our survey.

| Acronym | Full Name | Paper |
|---------|-----------|-------|
| LSTM | Long short-term memory | Hochreiter & Schmidhuber (1997) |
| CNN | Convolutional Neural Networks | LeCun et al. (1998) |
| Bi-LSTM | Bidirectional Long Short-Term Memory | Graves & Schmidhuber (2005) |
| Word2Vec | – | Mikolov et al. (2013) |
| GRU | Gated Recurrent Units | Cho et al. (2014) |
| Bi-GRU | Bidirectional Gated Recurrent Units | Cho et al. (2014) |
| DUAL-VGG | Dual Visual Geometry Group | Simonyan & Zisserman (2015) |
| FastText | – | Joulin et al. (2017) |
| BERT | Bidirectional Encoder Representations from Transformers | Devlin et al. (2019) |
| RoBERTa | Robustly Optimized BERT Pre-training Approach | Liu et al. (2019) |
| SciBERT | Scientific BERT | Beltagy et al. (2019) |

### 2.2 Patent Dataset and Repositories

Patent data are publicly available for bulk download from several sources in various formats such as XML, TSV, TIFF, and PDF. Examples include the USPTO, PatentsView[4], EPO[5], and WIPO[6]. Beyond these resources, several patent datasets are available for benchmarking purposes. The datasets are detailed in Table 3.

## 3 Methods

There has been a surge in research interest in developing AI-based methods in patent analysis. We organize the popular and important patent tasks that can benefit from AI tools. An overview of the major categorization of the patent tasks is shown in Figure 2.

---

[4] https://patentsview.org/
[5] https://www.epo.org/
[6] https://www.wipo.int/classifications/ipc/en/ITsupport/

Table 3: Overview of Patent Datasets: size, format, data type and intended tasks

| Dataset | Size | Format | Data type | Task |
|---|---|---|---|---|
| USPTO-2M Li et al. (2018) | 2M | JSON | text | Classification |
| BIGPATENT Sharma et al. (2019) | 1.3M | JSON | text | Summarization |
| USPTO-3M Lee & Hsiang (2020a) | 3M | SQL statement | text | Classification |
| PatentMatch Risch et al. (2021) | 6.3M | JSON | text | Retrieval |
| DeepPatent Kucer et al. (2022) | 350K | XML & PNG | text & image | Retrieval |
| DeepPatent2 Ajayi et al. (2023) | 2M | JSON & PNG | text & image | Retrieval |
| HUPD Suzgun et al. (2024) | 4.5M | JSON | text | Multi-purpose |

## 3.1 Classification

One of the major tasks of a patent reviewer is to assign a CPC or IPC code to the submitted patent. This task is time consuming due to the number of classification codes and their level of hierarchy. In the literature, several models have been used to automate this process. We organize them based on the nature of the method into three major categories. Table 4 represents a summary of the methods for patent classification.

Table 4: Summary of the papers for the patent classification task. Hierarchy levels for classification include Section, Class (white), Subclass (blue), Group, and Subgroup (grey). An example label "A61B 5/02" represents Section A, Class 61, Subclass B, Group 5, and Subgroup 02. The color green represents category of visualizations. The WIPO-alpha is a dataset for automated patent classification systems, and ALTA2018 is a dataset from Language Technology Programming Competition.

| Papers | Embeddings | Methods | Components |
|---|---|---|---|
| Grawe et al. (2017) | Word2Vec | Single layer LSTM | Description |
| Shalaby et al. (2018) | Fixed Hierarchy Vectors | LSTM | ADC |
| Risch & Krestel (2018) | FastText | GRU | Full text |
| Benites et al. (2018) | TF-IDF | SVM | Single Text Block |
| Risch & Krestel (2019) | FastText | GRU | Full text |
| Lee & Hsiang (2020a) | – | BERT-base | Claim |
| Althammer et al. (2021) | – | BERT, SciBERT | Claim |
| Sofean (2021) | Word2Vec | Multiple LSTMs | Description |
| Roudsari et al. (2022) | Word2Vec, FastText | BERT, XLNet, RoBERTa | Title, abstract |
| Kamateri et al. (2022) | FastText, Glove, Word2Vec | CNN, LSTM, GRU | TADC |
| Ghauri et al. (2023) | Vision Transformer | MLP | Image |
| Kamateri et al. (2023) | FastText | Bi-LSTM, Bi-GRU, LSTM | Metadata |
| Bekamiri et al. (2024) | SBERT | KNN | Claim, title, abstract |

Table 5: Existing results on the patent classification task. Hierarchy levels for classification include Section, Class, Subclass , Group, and Subgroup. The tuple (Result 1, Reuslt 2) denotes the results using (Data 1, Data 2) for the papers that report the measures using multiple datasets separately.

| Papers | Hierarchy Level | Accuracy | Precision | Recall | F1 | Top-3 | Data |
|---|---|---|---|---|---|---|---|
| Grawe et al. (2017) | Subgroup | 0.63 | 0.63 | 0.66 | 0.62 | – | USPTO |
| Shalaby et al. (2018) | Subclass | – | – | – | 0.61 | 0.79: F1 | - |
| Shalaby et al. (2018) | Class | – | – | – | 0.72 | 0.89: F1 | - |
| Risch & Krestel (2018) | Subclass | – | (0.49, 0.53) | – | – | (0.72,0.75): Precision | WIPO-alpha, USPTO |
| Benites et al. (2018) | Class | – | – | – | 0.78 | – | ALTA2018, WIPO |
| Risch & Krestel (2019) | Subclass | – | (0.49, 0.53) | – | – | (0.72,0.75): Precision | WIPO-alpha, USPTO |
| Lee & Hsiang (2020a) | Subclass | – | 0.81 | 0.55 | 0.65 | 0.44: F1 | USPTO |
| Althammer et al. (2021) | Subclass | (0.58, 0.59) | (0.57,0.58) | (0.58,0.59) | (0.56, 0.58) | – | USPTO |
| Sofean (2021) | Subclass | 0.74 | 0.92 | 0.63 | 0.75 | – | EPO, WIPO |
| Roudsari et al. (2022) | Subclass | – | (0.82, 0.82) | (0.55, 0.67) | (0.63, 0.72) | – | USPTO, CLEF-IP 2011 |
| Kamateri et al. (2022) | Subclass | 0.64 | – | – | – | – | CLEF-IP 2011 |
| Ghauri et al. (2023) | Image type | 0.85 | – | – | – | – | CLEF-IP 2011, USPTO |
| Kamateri et al. (2023) | Subclass | 0.68 | – | – | – | 0.89: accuracy | CLEFIP-0.54M |
| Bekamiri et al. (2024) | Subclass | – | 0.67 | 0.71 | 0.66 | – | USPTO |

### 3.1.1 Traditional Neural Networks

The commonality among these methods is that they follow a two-step approach: generate initial features and then use a classifier for the final classification. One of the initial studies, Grawe et al. (2017) implements a single-layer LSTM to classify patents at the IPC subgroup level where the initial features are obtained by the Word2Vec method. Similarly, Shalaby et al. (2018) use LSTM for IPC subclass level classification. For the initial document representation, the method uses fixed hierarchy vectors that utilize distinct models for various segments of the document. Risch & Krestel (2018) and Risch & Krestel (2019) focus on training fastText word embeddings on a corpus of 5 million patent documents and, then use Bi-GRU for classification. Similarly, Sofean (2021) apply text mining techniques to extract key sections from patents, train Word2Vec, and then use multiple parallel LSTMs for the classification task. These collectively show the usefulness of neural networks in patent analysis.

### 3.1.2 Ensemble Models

The models in this category use ensembling different word embeddings and deep learning models. Benites et al. (2018) use SVM as a baseline method and experiment with various datasets, the number of features, and semi-supervised learning approaches. Meanwhile, Kamateri et al. (2023) and Kamateri et al. (2022) both investigate ensemble models incorporating Bi-LSTM, Bi-GRU, LSTM, and GRU. More specifically, Kamateri et al. (2022) experiment with different word embedding techniques, whereas Kamateri et al. (2023) focus on applying various partitioning techniques to enhance the performance of the proposed framework. While the above methods heavily focus on texts, Ghauri et al. (2023) classify patent images into distinct types of visualizations, such as graphs, block circuits, flowcharts, and technical drawings, along with various perspectives including side, top, left, and perspective views. The approach utilizes the CLIP model with Multi-layer Perceptron (MLP) and various CNN models.

### 3.1.3 Pre-trained Language Models (PLMs)

The first study Lee & Hsiang (2020a) which involves PLMs, fine-tune the BERT model on the USPTO-2M dataset and introduces a new dataset, USPTO-3M at the subclass level to aid in future research. Concurrently, Roudsari et al. (2022) also fine-tune BERT, along with XLNet Yang et al. (2019), and RoBERTa on the USPTO-2M dataset. They establish XLNet as the new state-of-the-art in classification performance, achieving the highest precision, recall, and F1 measure. Althammer et al. (2021) implement domain adaptive pre-trained Linguistically Informed Masking and show that SciBERT based representations perform better than BERT-based representations in patent classification. SciBERT is pre-trained on scientific literature which helps the method to understand the technical language of patents. Bekamiri et al. (2024) use Sentence BERT which takes into account entire sentence instead of word by word. On USPTO data, their method gives the highest recall and F1 score.

### 3.1.4 Discussion and Suggestion

The evaluation measures for patent classification are accuracy, precision, recall, and the F1 score on the CPC or IPC. The earlier works on patent classification are mostly focused on simpler neural networks Grawe et al. (2017); Shalaby et al. (2018); Risch & Krestel (2018; 2019). Applying models such as LSTM can capture the sequence and context in the text which are suitable for patent domain since the context is critical. However, these are comparatively simple models that might be limited to capture complex technical structure in patent documentation. This limitation is evident in the evaluation metrics; for instance, the highest accuracy at the subclass level is only 0.74. More advanced techniques have become popular over time, including the adoption of PLMs. PLMs could be powerful because of their pretraining step on a massive amount of data. Patent text is different from the usual text in scientific articles (e.g., research papers). Thus, fine-tuning PLMs on patent datasets might be able to address some of these concerns by providing context-aware representations for the patent domain. From Table 5, it is noted that the precision on USPTO data was initially 0.53 Risch & Krestel (2018). With the adoption of pre-trained language models like BERT and RoBERTa, this metric has significantly improved to 0.82 Roudsari et al. (2022). The language models used for classification tasks in the patent domain are generally simpler compared to advanced large language models such as GPT and

LLaMA. There is a significant gap between the AI practices in the patent domain and the existing advance implementation of sophisticated models.

Table 6: Works on Patent Retrieval: The papers have white, blue, and gray based on the data type of text, image, and both respectively. Freepatent and Findpatent are patent data websites, where Findpatent includes patents registered in Russia. WIPS is a patent information search system.

| Work | Method | Training |
|---|---|---|
| Kravets et al. (2017) | CNN | supervised |
| Kang et al. (2020) | BERT | pre-trained |
| Chen et al. (2020) | BiLSTM-CRF, BiGRU-HAN | supervised |
| Jiang et al. (2021) | DUAL-VGG | supervised |
| Setchi et al. (2021) | SVM, Naive Bayes, Random Forest, MLP | supervised |
| Pustu-Iren et al. (2021) | RoBERTa, CLIP | pre-trained |
| Siddharth et al. (2022) | Sentence-BERT, TransE | pre-trained, unsupervised |
| Kucer et al. (2022) | (ImageNet, Sketchy) ResNet50 | supervised, finetuned |
| Higuchi & Yanai (2023) | Deep Metric Learning | self-supervised |
| Higuchi et al. (2023) | InfoNCE and ArcFace | self-supervised |
| Lo et al. (2024) | ResNet50, EfficientNetB-0, ViT-B-32, SwinV2-B ,CLIP , BLIP-2, GPT-4V | pre-trained, supervised |

Table 7: Results of the papers for the Patent Retrieval task. Here, mAP denotes mean average precision.

| Work | Data type | Data | Accuracy (%) | Precision | Recall | F1 | mAP |
|---|---|---|---|---|---|---|---|
| Kravets et al. (2017) | image | Freepatent, Findpatent | 30 | – | – | – | – |
| Kang et al. (2020) | text | WIPS | – | 71.74 | 94.29 | 81.48 | – |
| Chen et al. (2020) | text | USPTO | – | 92.4 | 91.9 | 92.2 | – |
| Jiang et al. (2021) | image | CLEF-IP, USPTO | – | – | – | – | – |
| Setchi et al. (2021) | text | IPO | – | – | – | – | – |
| Pustu-Iren et al. (2021) | image+text | EPO | – | – | – | – | 0.715 |
| Siddharth et al. (2022) | text | USPTO | 70.2 | 65.9 | 81.2 | 72.6 | – |
| Kucer et al. (2022) | image | DeepPatent | 70.1 | – | – | – | 37.9 |
| Higuchi & Yanai (2023) | image | DeepPatent | – | – | – | – | 0.85 |
| Higuchi et al. (2023) | image | DeepPatent | – | – | – | – | 0.622 |
| Lo et al. (2024) | image | DeepPatent2 | – | – | – | – | 0.69 |

Table 8: Summary of the methods on patent quality analysis: "Many" includes Linear regression, Ridge regression, Random Forest, XGBoost, CNN, and LSTM. "APR" stands for the measures accuracy, precision, and recall. IncoPat is a global patent database. We denote Attribute Network Embedding, Attention-based Convolutional Neural Network, European Telecommunications Standards Institute, Derwent Innovation by ANE, ACNN, ETSI, and DI respectively.

| Papers | Indicators | Methods | Evaluation Metrics | Datasets |
|---|---|---|---|---|
| Lin et al. (2018) | Citations, meta features | ANE, ACNN | RMSE | USPTO, OECD |
| Trappey et al. (2019) | PCA | DNN | Accuracy | ETSI and DI |
| Hsu et al. (2020) | Investor reaction, citations | Many | MAE | Patentsview |
| Chung & Sohn (2020) | Abstract, claims, predefined | CNN, Bi-LSTM | Precision, recall | USPTO |
| Aristodemou (2021) | 12 patent indices | ANN | APR, F1, FNR, MAE | USPTO, OECD |
| Erdogan et al. (2022) | 9 patent indices | MLP | Accuracy, Kappa, MAE | USPTO |
| Li et al. (2022) | Maintenance period | BiLSTM-ATT-CRF | APR, F1 | IncoPat |
| Krant (2023) | Patent text | MSABERT | MSE | USPTO, OECD |

## 3.2 Retrieval

We divide the relevant studies into three parts based on the deployed model for the retrieval task. Table 6 provides a concise overview of studies for patent retrieval.

### 3.2.1   Traditional Machine Learning

Initial studies have used traditional machine learning methods for patent retrieval. Setchi et al. (2021) describe five technical requirements to investigate the feasibility of AI for the task. These requirements include query expansion and identification of semantically similar documents. The study uses machine learning algorithms such as SVMs, Naive Bayesian learning, decision tree induction, RF along with word embeddings to solve the prior art retrieval problem. Prior art usually implies the references which may be used to determine the novelty of a patent application. Patent data is searched through multiple resources and returns results based on the query and the database and these results need to be merged to create the final result. Stamatis et al. (2023) employ techniques such as random forest, Support Vector Regression, and Decision Trees to effectively merge the search findings.

### 3.2.2   Traditional Neural Networks

The methods based on neural networks have been popular in recent years for patent retrieval. Kravets et al. (2017), Jiang et al. (2021), and Kucer et al. (2022) implement CNN, DUAL-VGG, and ResNet, respectively, to retrieve patent images based on a query image. Chen et al. (2020) aim to solve entity identification and semantic relation extraction by BiLSTM-CRF Huang et al. (2015) and BiGRU-HAN Han et al. (2019) respectively.

### 3.2.3   Pre-trained Language Models (PLMs) & Multimodal Models (MMs)

PLMs are useful in many text-related tasks and patent retrieval is not an exception. Pustu-Iren et al. (2021) utilize CLIP for image embedding alongside RoBERTa for capturing textual features, and thus, enhances the search process by incorporating both visual and textual data. Kang et al. (2020) use the BERT language model which includes the combinations of title, abstract, and claim. Siddharth et al. (2022) incorporate Sentence-BERT Reimers & Gurevych (2019) for text embeddings as well as use the TransE method for the citation and inventor knowledge graph embeddings. They identify that the mean cosine similarity among the vector representations of the patents is effective in linking multiple existing patents to a target patent.

Among other techniques, Higuchi & Yanai (2023), Higuchi et al. (2023) employ a deep metric learning framework with cross-entropy methods such as InfoNCE Oord et al. (2018) and ArcFace Deng et al. (2019). Multimodal techniques have also been used in the information retrieval Pustu-Iren et al. (2021). Here, the visual features are extracted using vision transformers, while textual features are from sentence transformers. Lo et al. (2024) use distribution-aware contrastive loss to improve understanding of class and category information which achieves robust representations even for tail classes. For captioning, they employ open-source BLIP-2 and GPT-4V, a frozen text encoder from CLIP for text feature, and various visual encoder backbones including ViT variants, ResNet50, EfficientNetB-0, and SwinV2-B.

### 3.2.4   Discussion and Suggestion

Patent retrieval process involves several subtasks such as defining technical requirements and merging search outcomes from various databases. The proposed methods often use traditional machine learning techniques like SVM, Naive Bayes, Decision trees, etc. While the image retrieval methods apply a variety of CNNs to effectively handle and analyze the visual data, the text retrieval methods have shifted towards PLMs such as BERT for advanced linguistic analysis. Clearly, traditional machine learning techniques are limited to capture the complexity of both patent image and text. While CNNs are popular for image retrieval tasks, the question remains in their effectiveness of patent image retrieval as patent images are non-traditional and technical. On the other hand, utilizing Vision Transformer alongside RoBERTa, Sentence-BERT, TransE shows a multimodal approach might be more suitable for handling the multimodal (e.g., text, images) aspect of patents. Pustu-Iren et al. (2021) demonstrate that the image and text-based transformer models achieve the highest mean average precision in patent retrieval tasks.

### 3.3 Quality Analysis

We divide the studies into two categories based on the methods for quality analysis. A summary of research on patent quality analysis is given in Table 8.

#### 3.3.1 Traditional Neural Network

Erdogan et al. (2022) apply an MLP-based approach for quality analysis, utilizing nine indices such as claim counts, forward citations, backward citations, the patent family size to measure the value of a patent, etc. Li et al. (2022) classifies patents based on their maintenance period in four categories. This study implements a Bi-LSTM along with the attention mechanism and Conditional Random Field (CRF) to classify the quality of a patent during the initial stages of its life cycle. The use of Deep Neural Networks is seen in Trappey et al. (2019) where 11 indicators were considered. Hsu et al. (2020) predict forward citation and investor reaction to patent announcements implementing CNN-LSTM neural networks and various ML models. Chung & Sohn (2020), Lin et al. (2018) and Aristodemou (2021) apply a variety of neural networks such as CNN, Bi-LSTM, Attention-based CNN (ACNN), deep and wide Artificial Neural Networks (ANN), respectively.

#### 3.3.2 Pre-trained Language Models (PLMs)

Krant (2023) propose a variation of BERT (MSABERT) to assess patent value based entirely on the textual data and use the OECD Eurostat, O. (2005) quality indicators for evaluation. Building upon BERT, MSABERT is capable of processing the multi-section structure and longer texts of patent documents. The OECD index includes composite indicators and generality with other predominant indices.

**Discussion.** While numerous measures are commonly used in assessing the quality of a patent, the absence of universally accepted "gold standard" poses a challenge. Several indices have been considered for patent valuation, among which forward citations are directly associated with the value—both monetary and quality— of a patent. While applying different deep learning models has some success, the question of building a method to handle technical information, metadata, images together inside a patent document remains open. Though MSABERT on entire dataset will be computationally costly, building upon it might be useful for quality evaluation.

### 3.4 Generation

The generative models are becoming increasingly popular in many domains. The recent developments in AI also have led to the novel research area of generating patents and thus reducing significant human effort that is otherwise needed to write a long document. However, only a couple of studies have attempted to address this problem. Lee & Hsiang (2020b) implement GPT-2 Radford et al. (2019) models to generate the independent claims in patents. The researchers fine-tune the model on 555,890 patent claims of the granted utility patents in 2013 from USPTO. Providing a few words, the method generates the first independent claim of the patent. However, the study is limited towards providing quantitative metrics to evaluate the effectiveness or quality of the patent claims generated by the model. In a separate study, Lee (2020) utilizes an alternative methodology, focusing on personalized claim generation by fine-tuning a pre-trained GPT-2 model with inventor-centric data. They also provide a few words or context as input to generate claims. The measure of personalization in the generated claims has been assessed using a BERT model. The underlying hypothesis is that the generated patent claims would demonstrate greater relevance to the respective inventor.

#### 3.4.1 Discussion and Suggestion

First, the problem of patent generation has not been adequately addressed. While GPT-2 and BERT have been applied for this task, patents require precise terminology and often contain complex, interrelated concepts that extend beyond the token context window utilized by GPT-2 and BERT. Also, GPT-2 sometimes produces ambiguous or vague text which is not suitable for patents. We believe that more advanced models (e.g., GPT-3.5) along with a large dataset for fine-tuning could help to improve the quality of generated patents.

### 3.5 Others

In addition to the above-mentioned tasks, there are other interesting studies in the patent domain. Recent work focuses on patent infringement, such as Chi & Wang (2022) develop a model with different deep learning methods, such as CNN and LSTM, to predict the possibility of a patent application being granted and classify the reason for a failed application. Another work Choi et al. (2022) applied a transformer and a Graph Neural Network (GNN) on patent classification for patent landscaping. Zaini et al. (2022) present an unsupervised method to identify the correlations between patent classification codes and search keywords using PCA and k-means. These studies provide advanced deep learning methods to avoid the risks in patent application. Moreover, there are various studies on generating new ideas and evaluating novelty, such as identifying the inventive process of novel patent using BERT Giordano et al. (2023), and an explainable AI (XAI) model for novelty analysis via Jang et al. (2023). Zou et al. (2023) propose a new task to predict the trends of patents for the companies, and also provide a solution for the task by training an event-based GNN. These studies bring new insights and directions for patent ideas and developments.

**Applications in Businesses.** The use of AI among businesses for patent related processes has significantly risen over time. Machine learning techniques are often being incorporated in 40% of all AI-associated patents reviewed. Additionally, the usage of the machine learning methods for these patents is growing at an impressive average annual rate of 28%[7]. Businesses are increasingly applying AI to enhance various aspects of the patent process, from drafting and classification to search and analysis. Some of the prominent examples include Qatent (2024), DaVinci (2024), and Questel (2024). Qatent (2024) leverages the latest NLP techniques to facilitate patent drafting for patent practitioners. It focuses on automating routine tasks—typing, automating renumbering of claims, and antecedence checking. It recommends various word and sentence alternatives during the claim drafting process, such as synonyms, broader or more specific terms, and other linguistic variations. Despite recent discussions around AI-generated inventions, Qatent maintains a human-centric approach which ensures all outputs are driven and controlled by human drafters. DaVinci (2024) is an advanced tool for drafting patents that uses generative AI to streamline the process. It supports a variety of document formats and lets users alter the AI's writing style to suit their needs. Questel (2024) offers AI powered patent classification, comprehensive patent search capabilities, efficient exploration of new markets, and opportunities such as management of patent fees and renewals.

**Language Models for Patents.** The trend of using language models (LM) in patent analysis is on the rise and involves both pretrained language models (PLMs) and large language models (LLMs). In classification tasks, text embeddings are often obtained from by BERT, SciBERT, RoBERTa, and Sentence-BERT, etc. In retrieval tasks, the application of BERT-based models for solving text-based retrieval is more common. However, so far, LMMs are not being used heavily in either of these tasks. For patent generation, LLM is being used, but more recent and advanced models like GPT-3 and its successor, LLama, and T5 have not been explored. Table 9 summarizes the methods with language models, the type of LLMs, and the corresponding patent tasks that are being addressed. In both classification and retrieval tasks, pretrained methods significantly improve the performance. For example, PLMs achieve the highest recall and F1 scores in the USPTO dataset. In patent retrieval, the use of different datasets (e.g., IPO, DeepPatent, EPO, WIPS, etc.), modalities (e.g., text, image, both text and image), and metrics (e.g., accuracy, mAP, precision, recall, F1) make it difficult to compare the results with other existing works.

## 4 Future Directions

Many researchers have employed AI to address a wide range of patent-related challenges. In spite of that, this field presents ample opportunities for further research. We strongly believe that a foundation Large Language/Vision-Language Model for patent data will provide a better comprehensive understanding and further improvement in performance on different tasks. Furthermore, it is beneficial to utilize AI for generating novel patent ideas and solutions as well as evaluating patent applications and their quality. We introduce the potential future directions in detail as follows.

---

[7]https://ip.com/blog/can-ai-invent-independently-how-ai-is-changing-the-patent-industry/

Table 9: Example of the works that used LLMs to solve patent tasks

| Work | LLM type | Task |
|---|---|---|
| Lee & Hsiang (2020a) | BERT-base | Classification |
| Althammer et al. (2021) | BERT, SciBERT | Classification |
| Roudsari et al. (2022) | BERT, RoBERTa | Classification |
| Bekamiri et al. (2024) | Sentence-BERT | Classification |
| Kang et al. (2020) | BERT | Retrieval |
| Pustu-Iren et al. (2021) | RoBERTa | Retrieval |
| Siddharth et al. (2022) | Sentence-BERT | Retrieval |
| Lee & Hsiang (2020b) | GPT-2 | Generation |
| Lee (2020) | GPT-2 | Generation |

## 4.1 Multimodal Learning on Patents

The availability of multiple modalities (e.g., text, images) in patent documents offers a comprehensive understanding of the related patent tasks. For the various tasks in the patent domain that are mentioned earlier, there is a lack of comprehensive datasets covering multiple modalities. The field could be further advanced by compiling larger datasets that include both textual and visual information found in patents. One of the challenges is that the patent images are often more complex and use advanced domain related concepts compared to the natural (or RGB) images. Thus, it may be necessary to get the opinion of corresponding domain experts to verify and curate the patent datasets accordingly. Though patent data is public, they still need a significant amount of effort in pre-processing to be analyzed effectively. For instance, the images are low-quality sketches and will be difficult to be utilized for the patent classification task. The texts of design patents contain limited information. Introducing new multimodal datasets would be helpful for benchmarking in many fields beyond patent analysis such as question answering in vision tasks.

Most of the AI-based patent analysis methods have used either text or image data but not both of them. Recent advances in multimodal learning would allow for more reliable and accurate AI-driven patent analysis. For example, consider the simplest task of patent classification. Here, multiple studies have incorporated patent text and metadata to some extent Lee & Hsiang (2020a); Grawe et al. (2017). However, these studies have largely overlooked the potential of patent images. Intuitively, drawings or sketches provide geometrical information about individual patents. So for the classification task, images may be grounded with domain knowledge that is available in the form of text descriptions. In general, multimodal learning can be used to *align representations* derived from text descriptions with those derived from technical images. By doing so, we can guarantee that predictions (classification on unseen or test data) after training are more reliable for related tasks. As these methods will make predictions for important tasks, the question of generating explanations of those predictions by these models also becomes relevant.

## 4.2 Generative AI for Patents

The domain of patent generation remains relatively unexplored despite its importance. Generating novel and innovative patent claims, along with abstracts and other important sections of the patent document using different inputs, is an exciting potential research field. However, in patent generation, LLMs can suffer from hallucination, where they generate incorrect and fictitious information. They might produce repetitive and monotonous texts which will lack creativity. Further, to mitigate the risk of patent infringement, it is also crucial to feed LLMs with the up-to-date patent data. This will ensure that the generated text does not replicate any existing inventions. Thus, the generation process requires human oversight and feedback to ensure accuracy and relevance and cannot be fully automated yet. On the other hand, the assessment of the text generated by the generative models is also challenging. In the literature, machine-centric and token-based methods are used to evaluate generative language models Lee (2023). As patents include jargons and many domain specific words, evaluating generated patent text in terms of only natural language will not be sufficient. Thus, the important question becomes—*how to construct domain-specific evaluation measures for the synthetic or the generated text from LLMs?*

### 4.3 AI for Patent Assessment

To asses patent's novelty, one of the major tasks is to retrieve similar patents to determine whether the patent is significantly different from existing patents. One of the important task in this case is to generate search queries. This often needs alternate search terms, related words, and synonyms which require domain knowledge. The quality and structure of queries directly impact the relevance of the search results. The current AI methods are yet to automate this entire process. Thus, it brings challenges to obtain adequate similar patents and correctly assess patent's innovativeness and novelty. On the other hand, the generic quality analysis are based on well-known measures Aristodemou (2021); Erdogan et al. (2022). Nonetheless, it remains unclear which of these indices are associated with the actual value of the patent (e.g., generated revenue).

### 4.4 Building a Knowledge Graph

Patents are represented as nodes connected by edges such as citations in a citation network Liu & Li (2022). This structured representation allows for detailed citation analysis which is considered a crucial metric in understanding a patent's value. One interesting future direction would be to build a knowledge graph using other important information such as meta-data, semantic similarity of patents, etc. This may lead to a more organized landscape of patents. This knowledge graph can help with prior art searches, the identification of related patents, and identify valuable patents (e.g., patents with high citations) Siddharth et al. (2022).

## 5   Conclusions

In this survey, we have provided a comprehensive overview of various patent analysis tasks with AI models. We have presented a novel schema with a detailed organization of the research papers, analyzing the methodologies used, their advantages, limitations, and how they apply to patents. The process from filing a patent application to being it granted often requires a considerable amount of time. Given the crucial role of patents in economic development and the lengthy process from application to grant, the need for AI-driven approaches in this field is growing to automate different parts of this process. Although numerous studies have addressed this issue, the modern AI techniques have opened many directions for improvement. We have offered several insights into such potential future directions. This survey aims to be a useful guide for researchers, practitioners, and patent offices in the intersection of AI and patent systems.

**Broader Impact: Benefits and Limitations.** The life-cycle of a patent—the time from its submission to acceptance—is lengthy as it undergoes significant scrutiny and multiple iterations of revisions. The advancements in AI can make this process faster and thus, can essentially accelerate technological innovation. For instance, while reviewing, AI tools can help retrieve relevant documents more efficiently and accurately than a human reviewer who often requires enough experience.
There are also a few limitations of using AI in patent analysis. First, the AI methods may lack the nuanced understanding that human experts possess. Second, evaluation scores for AI methods in classification and retrieval indicate lower accuracy, (see Tables 5 and 7) and thus they still need human intervention to obtain relevant literature—which is important while reviewing—to prevent the patent infringement issues. Therefore, the entire process cannot be fully automated, and it is important to have human experts in the loop. This requirement also applies to generative models for patent drafting (Sec. 3.4) which needs human guidance for accuracy. Additionally, there are ethical concerns regarding the potential displacement of human workers by AI tools.
*The Impact of Existing Works.* Some of the major benefits are as follows: (1) Speed: The inclusion of AI in patent analysis tasks will speed up the review process. For example, Ghauri et al. (2023) use a vision transformer that classifies images much more efficiently than previous works, and Bekamiri et al. (2024) achieve higher recall in classification tasks. Since patent classification is a time-consuming task for a human expert, incorporating these advancements into the review process will make the process faster. (2) Novelty: Another important task is retrieving similar patents which is essential to assess the novelty of a patent. Higuchi & Yanai (2023) show a satisfactory mAP in retrieving similar images, which can play a key role in patent infringement. (3)Innovation: Lee & Hsiang (2020b); Lee (2020) explore generating new patents,

which is an important component to foster new innovation. This research provides inspiration for further development in the field including creation of new and innovative patents.

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
