# OpenReview forum: "A Comprehensive Survey on AI-based Methods for Patents"
_TMLR — Rejected by TMLR_

### Review · Reviewer_7JES · 2024-06-06

**Summary Of Contributions:**

This manuscript provides a systematical review for AI-based methods used in the patent analysis. Specifically, the authors focus on four aspects of patent analysis, namely classification, retrieval, generation and quality assessment, with rigorous mathematical formulation and detailed challenges discussion. This could be a comprehensive review of the specific application of AI-based methods in the subfield of patent analysis.

**Audience:**

Yes

**Broader Impact Concerns:**

As this manuscript is discussing potential impact of the AI-based method on specific real world applications, I think it would be better to include some broader impact discussion on how these AI-based methods can benefit and hurt the stuffs related to the patent.

**Claims And Evidence:**

No

**Requested Changes:**

Can you have some additional discussion on the significance and the limitations of the methods mentioned in the paper? Also, some discussions on the gap between those demonstrations and practices will be helpful.

**Strengths And Weaknesses:**

* Strengths: A comprehensive overview.
* Weaknesses: This review focus on the qualitative side with little efforts on quantitative side. Hence, it can be hard to know the benefits and drawback of each individual method.

---

> ### Author Response · Authors · 2024-09-07
> **Rebuttal**
>
> Thank you for the comments. Please see the response below. The revision in the draft is in blue.
>
> **1: Can you have some additional discussion on the significance and the limitations of the methods mentioned in the paper? Also, some discussions on the gap between those demonstrations and practices will be helpful.**
>
> **Response:** We have incorporated additional discussion and suggestions on the significance and limitations of the methods. These are presented in **Sections 3.1.4** and **3.2.4**.
>
> **2: As this manuscript is discussing potential impact of the AI-based method on specific real world applications, I think it would be better to include some broader impact discussion on how these AI-based methods can benefit and hurt the stuffs related to the patent.**
>
> **Response:** Thanks for the suggestion. We have included **‘Broader Impact: Benefits and Limitations’** paragraph in the conclusion. In this section, we discuss the advantages of incorporating AI-based methods into patent analysis in general and the potential negative impacts of these methods. We specifically mention the need for human invovement in deploying AI-based models.

---

### Review · Reviewer_uvHJ · 2024-06-24

**Summary Of Contributions:**

This paper provided a comprehensive overview of machine-learning methods in patent analyses. The authors reviewed forty related publications and applications from 2017 to 2023. The paper first presented a taxonomy of the patent analysis tasks, splitting them into four groups: classification, retrieval, quality analysis, and generation. Each group was further divided into subgroups of methods. Each subgroup was summarized and discussed. In the classification, the authors discussed that fine-tuning LLMs has the potential to gain higher performance. In the retrieval, the authors suggested the multi-modal approaches. In the quality analysis and generation, the authors pointed out that there was a lack of gold standards and measurements. Finally, the authors offered four directions for future work addressing the previously discussed issues.

**Audience:**

Yes

**Broader Impact Concerns:**

This is a survey paper. No ethical concerns.

**Claims And Evidence:**

No

**Requested Changes:**

## Major changes

1. Provide a more in-depth analysis of the conclusions and suggestions. I believed it was important to support them with existing work in the survey (the forty papers) or other similar tasks.
2. Discuss why the taxonomy was helpful and showed unique insights compared to the previous survey.
3. Provide details of the search and inclusion criteria.

## Minor changes

1. Change LLMs to PLMs (pre-trained language models) or transformer-based pre-trained language models.
2. When presenting performance, it would be more insightful to see the datasets and the performance in one table.
3. Section 4.3, "patent assessment," was out-of-context. The paragraph discussed the retrieval and quality analysis tasks.

**Strengths And Weaknesses:**

## Strengths

1. The paper provided an updated summary of the patent analysis tasks. This paper included publications from 2017 to 2023, whereas the previous survey paper (Krestel et al., 2021) only included publications up to 2020.
2. The paper provided evaluation metrics of the reviewed papers.

## Weaknesses:

1. This paper's analysis could be more profound. While it discussed subgroups of each task, it did not provide a relationship between facts to support its suggestions. For example, in the discussion of the classification task, the authors suggested fine-tuning LLMs but did not discuss trends, as shown in Table 5, to support its suggestion. Similarly, when the authors suggested a combination of Vision Transformer and RoBERTa for the retrieval tasks, they did not discuss any evidence to support it.  In addition, the discussion often did not discuss how the challenges mentioned in Section 2 had been studied or solved by the reviewed publications. Another example was the lack of support in the knowledge graph suggestion (Section 4.4).
2. The proposed taxonomy was somewhat incomplete. Although the most recent survey paper (Krestel et al., 2021) did not provide a taxonomy, it did introduce groups of patent analysis tasks. But there was no discussion on why the proposed taxonomy was better (i.e., would you still get the lack of gold standards if you follow the division in Krestel et al., 2021?) In addition, BERT and similar models might not be qualified as LLMs in the taxonomy.
3. There was no detail on how the authors included the publications in the survey. While this was not a systematic literature review, it would be helpful if the authors provided the details of the methodology used in searching and including the publications in the paper. This could provide meaningful insights towards the comprehensive of the survey.

---

> ### Author Response · Authors · 2024-09-07
> **Rebuttal**
>
> Thank you for the comments. Please see the response below. The revision in the draft is in blue.
>
> **1: Provide a more in-depth analysis of the conclusions and suggestions. I believed it was important to support them with existing work in the survey (the forty papers) or other similar tasks**
>
> **Response:** Thank you for this comment. As suggested, we have integrated these into our discussions and suggestions (Sections **3.1.4** and **3.2.4**) to substantiate the suggestions with existing works.
>
> **2: Discuss why the taxonomy was helpful and showed unique insights compared to the previous survey.**
>
> **Response:** Our taxonomy is unique in a few ways. First, our taxonomy is based on the AI tools and patent-related tasks. This provides an in-depth view of the methods being used in specific tasks. Additionally, it organizes the previous works based on the type of the methods (e.g., Neural networks, Pre-trained language models, etc.) and provides granularity. This will be also beneficial for future researchers who will aim to focus on a task-specific method. Also note that, earlier studies do not offer this type of taxonomy that directly narrows down to specific tasks and methods. Second, another uniqueness about this organization is that it captures the recent trends of using advanced methods (e.g., LLMs) which is missing in the existing survey.
>
> We incorporated the above mentioned paragraph in **introduction**.
>
>
>
> **3:Provide details of the search and inclusion criteria.**
>
> **Response:** We have conducted our literature search using Google Scholar and Semantic Scholar, focusing on various categories of patent-related tasks and the application of AI methods. To align with the recent trends, we have limited our search to publications from 2017 to 2024. Our search criteria included various keywords such as 'patent', 'AI in patent', 'patent classification', 'patent tasks', 'patent retrieval', 'survey in patent', 'patent generation', 'patent quality analysis', and 'patent dataset'. This combination of search terms has yielded hundreds of patent-related research papers. We have excluded more than half of these papers after reviewing their titles and abstracts, as they have not met our criteria (e.g., they did not fall under any of the relevant categories). For instance, one retrieved paper titled 'Generating patent development maps for technology monitoring using semantic patent-topic analysis' is not relevant to this survey. After thorough scrutiny and reorganization, we have included more than 40 papers for the survey.
>
> We included this paragraph in **Section 1.1 (Overview)**.
>
> **Minor changes**
>
> **1:Change LLMs to PLMs (pre-trained language models) or transformer-based pre-trained language models.**
>
> **Response:** We changed all the LLMs to Pre-trained language models (PLMs). We have also made the changes in **Figure 1**.
>
> **2:When presenting performance, it would be more insightful to see the datasets and the performance in one table.**
>
> **Response:** This is a nice suggestion. We have made the changes in **Table 5** and **Table 7** where we have included one column for the datasets.
>
> **3: Section 4.3, "patent assessment," was out-of-context. The paragraph discussed the retrieval and quality analysis tasks.**
>
> **Response:** We apologize for the confusion. To asses patent's novelty one of the major tasks is to retrieve the similar patents to determine whether the patent is significantly different from the existing works. We have re-written **Section 4.3** and clarified these accordingly.

---

### Review · Reviewer_EWLm · 2024-08-29

**Summary Of Contributions:**

The paper provides a survey of AI-based methods for patent analysis. It covers recent AI tools applied to different tasks and introduces a novel taxonomy for categorizing these tools based on the tasks in the patent lifecycle and AI methods. The paper also offers insights into current trends and future research directions.

**Audience:**

Yes

**Broader Impact Concerns:**

Given the significant influence that AI can have on the innovation process and intellectual property rights, it is crucial to consider issues such as algorithmic bias, the transparency of AI-driven decisions, and related concerns. The paper should include discussions about it.

**Claims And Evidence:**

Yes

**Requested Changes:**

1.	LLMs is a separated category of AI methods for different tasks, and it is appeared to be the future trend. The authors should discuss more about the differences in task formulation and improvements when applying LLMs to tasks compared to traditional ML methods. (important)
2.	Although the different tasks and AI methods are categorized nicely, the timeline is unclear. The authors should also outline the trends in the types of tasks researchers are interested in and the trends in AI methods. (important)
3.	If applicable, the authors could include broader impact and ethical concerns related to the works mentioned in the paper. (beneficial)
4.	The paper could benefit from discussions about the societal impact and ethical implications of using AI for patent analysis overall. (important)

**Strengths And Weaknesses:**

The survey is well-written, with comprehensive figures and tables. The introduction of the taxonomy for categorizing the works by tasks and AI methods is a highlight of the paper, offering a structured way to approach the diverse works in this area. The paper also effectively outlines several interesting potential future research directions.
While the survey is broad, the depth of the analysis for each method could be enhanced, particularly in comparing the effectiveness and limitations of the different AI methods. The paper also lacks discussion on the broader impact and ethical implications of AI in patent analysis, which is increasingly important nowadays.

---

> ### Author Response · Authors · 2024-09-07
> **Rebuttal**
>
> Thank you for the comments. Please see the response below. The revision in the draft is in blue.
>
> **1:LLMs is a separated category of AI methods for different tasks, and it is appeared to be the future trend. The authors should discuss more about the differences in task formulation and improvements when applying LLMs to tasks compared to traditional ML methods. (important)**
>
> **Response:** Thank you for the suggestion. While the task formulations remain same, the recent methods with LLMs use different datasets. This makes it difficult to have a fair comparison with the existing techniques. Nonetheless, we have added a discussion in the last paragraph in Section 3.5 and summarized the methods in Table 9.
>
>
> **2:Although the different tasks and AI methods are categorized nicely, the timeline is unclear. The authors should also outline the trends in the types of tasks researchers are interested in and the trends in AI methods. (important)**
>
> **Response:** Thank you for the comment. Please note that the tables are organized in chronological order to show how over the years the applied methods have been evolved. For example, in patent classification (see. **Table 4**), the initial methods are simpler embedding techniques such as Word2Vec, TF-IDF, etc. Over time, more advanced methods such as Vision Transformers and pretrained language models (PLMs) including BERT and their variations have been deployed.
>
> Our survey focuses on the important tasks in patent analysis and we outline the trends in individual tasks (e.g., classification, retrieval, generation). The researchers are incorporating PLMs (e.g., BERT and RoBERTa), LLMs (e.g., GPT), Vision transformers (e.g., ViT and Swin)  and multimodal models (e.g., CLIP and BLIP) for all these tasks. In the discussion and suggestion sections (**Sections 3.1.4, 3.2.4, 3.4.1**), we discuss this shift towards the advanced methods and their significance in details.
>
>
>
> **3:If applicable, the authors could include broader impact and ethical concerns related to the works mentioned in the paper. (beneficial)**
>
> **Response:** We understand the next question is similar to this one. Please see our response for the next question. According to your suggestion, we have also added a discussion about the positive impact of the existing papers. The negative impact has been discussed  mainly about using AI for patent analysis and they are also relevant for the existing works.
>
> **4:The paper could benefit from discussions about the societal impact and ethical implications of using AI for patent analysis overall. (important)**
>
> **Response:** Thank you for the suggestion. A paragraph named **'Broader Impact: Benefits and Limitations’** is added in the revision (**Section 5**). We mainly discuss how AI based patent analysis can benefit the process (Positive societal impact) and the limitations that can lead to patent infringement (Negative societal impact). We also discuss ethical issues.

---

### Author Response · Authors · 2024-09-07
**Thank you**

We thank all the reviewers for their insightful suggestions. We addressed the suggestions in our revised version. In the draft, the revision is highlighted in blue. A detailed point-wise rebuttal is presented below.

---

### Decision · Action_Editor_y2Re · 2024-10-10

**Recommendation:** Reject

**Comment:**

The paper aims to survey an interesting topic on AI for patent applications. However, the current version of the work is not ready for publications. More work should be surveyed and the suggestions/argument should be better back-up by experiments and/or previous work.

**Audience:**

The paper is of interest to a small community working at the intersection of AI and patent.

**Claims And Evidence:**

The paper aims to provide a survey on AI solutions for patent analysis. However, the claims are not well-supported by evidence. As pointed out in the reviews, some suggestions such as "use advanced large language models for the classification tasks " are not back-up with experiments or references. In addition, the paper discusses a very limited subset of the work on patent analysis between 2017 and 2023.